# Assessment of Miscanthus Yield Potential from Strip-Mined Lands (SML) and Its Impacts on Stream Water Quality

**Kamalakanta Sahoo** [1,2,*] **, Adam M. Milewski** [3] **, Sudhagar Mani** [1] **, Nahal Hoghooghi** [4] **and Sudhanshu Sekhar Panda** [5]

1 School of Chemical, Materials and Biomedical Engineering, University of Georgia, Athens, GA 30602, USA; smani@engr.uga.edu
2 Forest Products Laboratory, United States Forest Service, Madison, WI 53726, USA
3 Department of Geology, University of Georgia, Water Resources and Remote Sensing Laboratory (WRRSL), Athens, GA 30602, USA; milewski@uga.edu
4 School of Environmental, Civil, Agricultural, and Mechanical Engineering, University of Georgia, Athens, GA 30602, USA; nahalh@uga.edu
5 Institute of Environmental Spatial Analysis, University of North Georgia, Oakwood, GA 30566, USA; Sudhanshu.Panda@ung.edu
* Correspondence: sahoo@uga.edu; Tel.: +1-706-351-1037

**Abstract:** Strip-mined land (SML) disturbed by coal mining is a non-crop land resource that can be utilized to cultivate high-yielding energy crops such as miscanthus for bioenergy applications. However, the biomass yield potential, annual availability, and environmental impacts of growing energy crops in SML are less understood. In this study, we estimated the yield potential of miscanthus (*Miscanthus sinensis*) in SML and its environmental impacts on local streams using the Soil and Water Assessment Tool (SWAT). After calibration and validation of the SWAT model, the results demonstrated that miscanthus yield potentials were 2.6 (0.8−5.53), 10.0 (1.3−16.0), and 16.0 (1.34−26.0) Mg ha$^{-1}$ with fertilizer application rates of 0, 100, and 200 kg-N ha$^{-1}$, respectively. Furthermore, cultivation of miscanthus in SML has the potential to reduce sediment (~20%) and nitrate (2.5−10.0%) loads reaching water streams, with a marginal increase in phosphorus load. The available SML in the United States could produce about 10 to 16 dry Tg of biomass per year without negatively impacting the water quality. In conclusion, SML can provide a unique opportunity to produce biomass for bioenergy applications, while improving stream water quality in a highly dense mining area (the Appalachian region) in the United States.

**Keywords:** strip-mined land; bioenergy; biomass; energy crop; miscanthus; SWAT model; SWAT-CUP; runoff; nutrients; and water quality

## 1. Introduction

Biomass is one of the important low-carbon renewable resources for creating a sustainable bio-economy and can be attractive for mitigating global warming, providing energy security, and creating jobs [1–3]. A large quantity of biomass is required to meet the United States' biofuel production mandate, the Energy Independence and Security Act (EISA), 2007, i.e., 60 billion liters of cellulosic liquid biofuel [4]. In addition, a large quantity of biomass is needed to produce renewable electricity to meet the Renewable Portfolio Standard (RPS) mandate [5], bioproducts, and industrial chemicals [6], which are currently produced from non-renewable fossil resources.

Perennial energy crops such as miscanthus and switchgrass have been proposed as suitable non-invasive plant species to produce biomass due to their high biomass yield, lower inputs requirement and higher nutrient use efficiency, and higher rates of carbon sequestration [7]. While energy crops have very high yield potentials, the large demand for biomass to meet the U.S. biofuel production mandate (EISA, 2007) would require a large cultivated land area (i.e., 15.8 Mha of land to produce 60 Gl of cellulosic biofuel, which is about 10% of total available cropland in the USA) [8] and growing energy crops on traditional croplands may displace food production, which may raise concerns such as the rise of food prices and insufficient food supply—Food Insecurity [9]—and other environmental problems such as deforestation [2]—indirect land use change (ILUC). Therefore, effective utilization of non-agricultural lands for biomass production without degrading its soil and water quality is critical for the long-term sustainability of bioenergy sector in the USA or elsewhere in the world.

In the USA, about 3.5 Mha [10,11] of strip-mined lands (SMLs) are available that could be used to cultivate energy crops while reducing the cost of reclamation, improving soil and water quality, and providing environmental and economic benefits to our society. Figure 1 shows the geographical distribution of coal mined lands across the USA. The Appalachian region covers most of the surface coal mining locations at various stages of land reclamation programs. Surface coal mining reclamation lands could be integrated with bioenergy programs to produce sustainable energy crops while restoring soil and water quality [12].

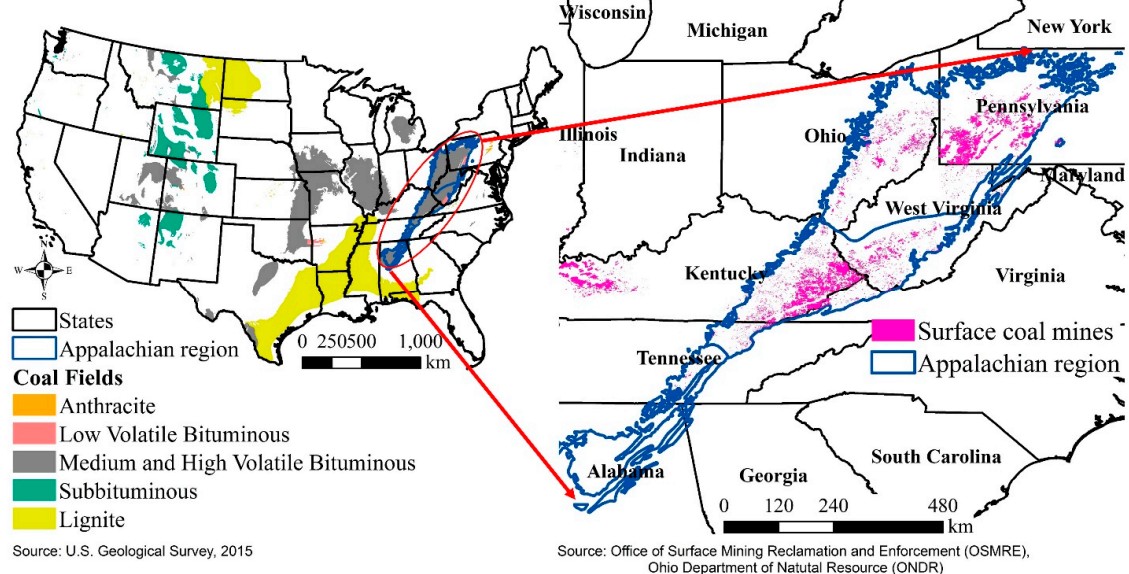

**Figure 1.** Geographical distribution of coal mined lands across the USA and the Appalachian region.

The recent experimental plot studies have demonstrated the potential of growing energy crops on marginal lands [7,13,14] and SMLs [10,15–17]. The experimental studies on the cultivation of miscanthus in SML have also demonstrated that the biomass yield varied widely from 4.9 to 21 dry Mg ha$^{-1}$ due to the variations in the management/agronomic practices, climate, and soil based on the recently established (3-4 years) miscanthus stands [16]. Similarly, switchgrass yields from SML also varied widely within the experimental plots due to the soil, climate, and land management practices [18]. Therefore, it is important to accommodate these variations while predicting biomass yield potential in a large study area/region.

Several empirical models have been developed based on the field study [8] and biophysical models such as WIMOVAC, MISCANMOD, etc. to predict miscanthus yield [19,20]. However, they did not assess the impacts of soil and water quality, and nutrient stress on biomass yield in SMLs or marginal lands [21]. SMLs are naturally very low in nutrients and are more susceptible to poor soil

and water quality [22,23]. For example, water holding capacity (3–33 cm$^3$ cm$^{-3}$), soil organic carbon content (7.8–20.6 g kg$^{-1}$), nitrogen (0.8–1.6 g kg$^{-1}$) and phosphorous (27.0–129.0 mg kg$^{-1}$) contents in SML soils are not sufficiently adequate to grow crops compared with that of agricultural cropland soils [22]. Therefore, assessing the soil and water quality impacts of strip-mined soils for energy crops cultivation is critical to establish a sustainable biorefinery in the USA.

The Soil and Water Assessment Tool (SWAT) is often used for predicting the impact of land management practices on runoff, biomass yield, and water quality parameters such as stream loading of sediment and nutrients [13,24]. It is a physically based, semi-distributed, and daily time-scale watershed hydrology model that uses data/information related to land use, soil, topography, climate, and land management practices [25]. It has also been widely used to predict the biomass yield of perennial energy crops on productive agricultural lands with nutrient-rich soil types [21,26]. Therefore, the watershed modeling approach using SWAT platform can provide the necessary knowledge and information to analyze the impact of growing energy crops on SML, while estimating biomass yield, and its production viability [27].

The objectives of this study were (i) to develop a watershed model to simulate the existing watershed characteristics and evaluate the hydrologic and environmental impacts of growing miscanthus in SMLs using SWAT, and (ii) to estimate the biomass yield potential and availability of miscanthus from SMLs in a large study area (the state of Ohio) for bioenergy production. This is the first study that used the SWAT model to analyze the impacts of growing energy crops in SML available in the USA and predicts its biomass yield with different management practices. The results produced in this study will provide new and helpful information to the scientific community and different stakeholders linked to SML in the USA as well as at the global level.

## 2. Methodology

The Soil and Water Assessment Tool (SWAT) was used to analyze the impact of growing miscanthus in strip-mined lands (SML) on water quality and biomass yield potentials. The major input data required to build a SWAT model include DEM (Digital Elevation Model), slope, soil properties, land use, crop management practices, and meteorological information. During watershed modeling, either Cropland Data Layer (CDL) or National Land Cover Database (NLCD) can be used as land use input data in the SWAT. However, the spatial extent of SMLs was not represented in these databases due to the limited amount of such lands. SMLs are included in other usual land classifications such as fallow land or developed land. Therefore, it is necessary to identify the spatial extent of strip-mined land in the land use dataset as a required input to the SWAT model.

*2.1. Identification of Strip-Mined Land and Its Spatial Extent*

2.1.1. Availability of Strip-Mined Land in the USA

Surface mining activities disturbed about 2.5 Mha of land between 1930 and 1977 and a large part of that area is with little or no reclamation [10]. The U.S. Government passed the Surface Mining Control and Reclamation Act (SMCRA) in 1977 to reduce environmental degradation and risks associated with mined lands [28]. According to this law, the disturbed land due to mining activities must be reclaimed and returned to its original condition otherwise the concerned party is liable for a penalty [29]. In addition to already-disturbed lands by mined activities before enacting SMCRA law, more than 1 Mha of the surface-mined/strip-mined area have been disturbed and reclaimed since the SMCRA was enacted [11]. Usually, these lands have been brought back to pre-mined conditions in phases that include contouring, replacing previously stripped topsoil, and revegetation [10]. Currently, about 2 Mha of these lands are still under license for mining [11].

We compiled information from OSMRE about the stock of surface mined land status in the USA (Table S1) and Ohio (Table S2) [11] and presented in the supporting document. It was estimated that about 0.9 and 0.05 Mha of reclaimed SML are available in the USA and Ohio, respectively. In addition,

annually about 37,000 and 2000 ha of strip-mined lands from the post-SMCRA law are added to the current SML inventory. Miscanthus is one the most productive and environmentally effective (i.e., high carbon sequestration, low nutrient requirement, high water use efficiency) among all energy crops in the USA [30]. It can help to reduce the high-cost SML surface reclamation by sharing part of the biomass production cost for bioenergy use [10,31].

### 2.1.2. Identification of SMLs Suitable for Miscanthus Cultivation

The polygon dataset for strip-mined land was acquired from the Ohio Department of Natural Resources (ODNR) to spatially locate and identify strip-mined lands available in Ohio [32]. The surface mined lands are categorized into A, B, C, and D groups depending on the time of awarding the mining permit and going from oldest to most recent, with A being the oldest group. Depending on the current mining activity, these lands are also classified as active (ACT), inactive (INA), temporarily inactive (TIN), released (REL), or abandoned (ABA).

In the state of Ohio, about 87,000 ha of lands were strip-mined before SMCRA, 1977. There are about 120,000, 60,500, 14,600, and 15,200 ha of the lands classified into A, B, C, and D groups of the SML permit categories, respectively. All A and B category surface mine lands are abandoned, and the inactive category includes C. About 60,250 ha of land are classified as inactive but are included in group C. There are about 6100, 30,500, 5900, and 74,850 ha of land categorized as abandoned, active, inactive, and released, respectively, in the D-group. Based on slope and soil type, the D-group SML in the released category is most suitable for growing energy crops (personal communication with geospatial data specialists at ODNR). The geographical spread of SML was in the eastern portion of Ohio and a large concentration of these lands falls in three counties (i.e., Belmont, Jefferson, and Harrison). The land cover dataset classification (i.e., CDL dataset) did not explicitly include the SML as a unique category—merged with other land classifications. Therefore, the SML category was appended to CDL land classification with the help of GIS tools before using it in the SWAT model.

### 2.2. Study Area Descriptions

The study area located in Ohio includes the Upper Ohio-Wheeling (USGS 8-digit HUC–05030106) and Tuscarawas watersheds (USGS 8-digit HUC–05040001). The total study area of the delineated cover is approximately 2632.5 km$^2$ spread over Belmont, Harrison, Guemsay, and Jefferson counties (Figure 2). The selected watershed drains in the Ohio River. The mean elevation of the watershed is 337 m (188–428 m). Land cover types include forest (58%), pasture (22%), developed area (8.5%), SML (8%), and others (4.5%) in the watershed. The majority of the land area is the silt loam soil type with more than 10% slope. The watershed received annual average precipitation of 1180 mm during the study period (2000–2013).

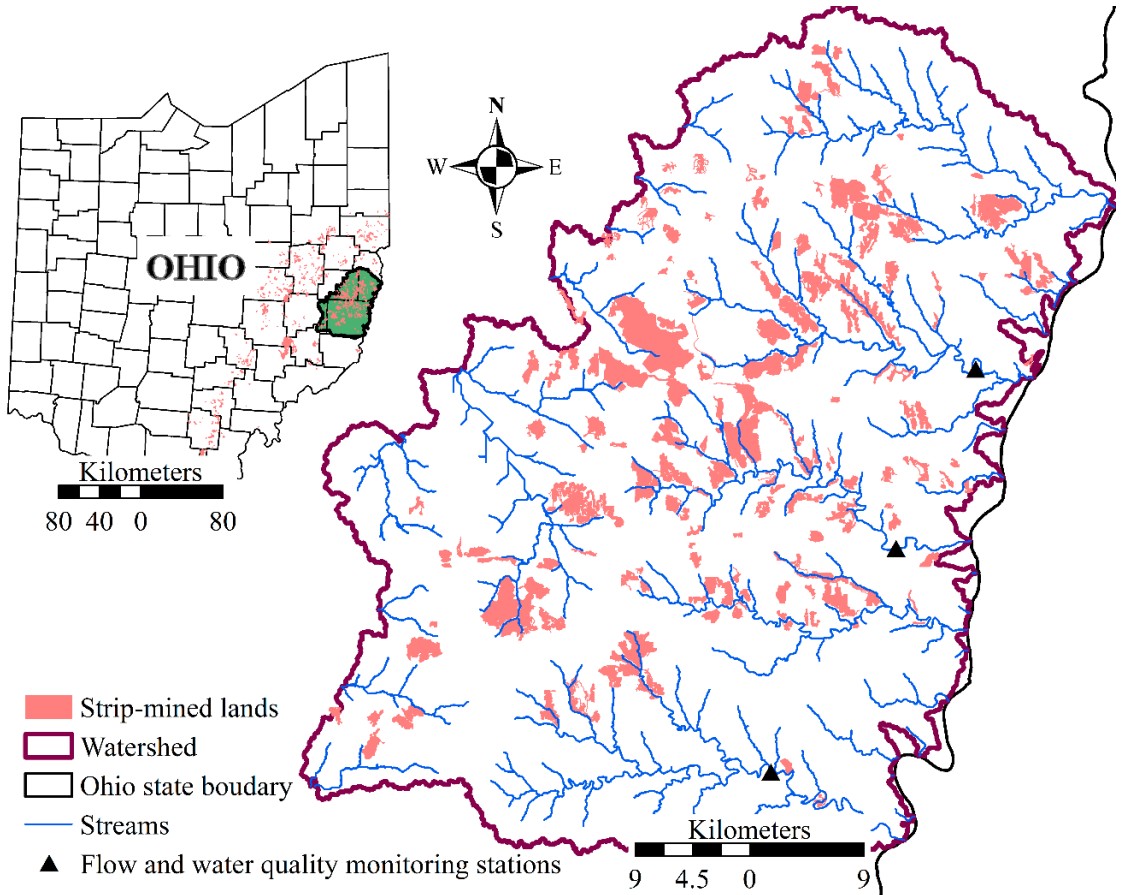

**Figure 2.** Delineation of studied watershed and streams in the state of Ohio with strip-mined land (SML).

*2.3. SWAT Model*

The SWAT is a physics-based, semi-distributed, and daily timescale watershed hydrology model and uses data/information related to land use, soil, topography, climate, and land management practices [25]. The SWAT model delineates watersheds, divides them into multiple sub-watersheds that are further subdivided into hydrologic response units (HRU). An HRU is a land unit having a unique land use, soil type, and slope. Flow, sediment, and nutrient yield across all HRUs in a subwatershed are summed and the resulting flow and loads are then routed through channels to the watershed outlets. The major components of SWAT include many integrated modules related to hydrology, climate, crop growth, nutrients, etc. [25,33]. The SWAT model can simulate the growth of a wide selection of crops using the Erosion-Productivity Impact Calculator (EPIC) model that calculates leaf area development, sunlight interception, and conversion to biomass, considering environmental constraints resulting from water, nutrients, and temperature [13,21,26]. SWAT quantifies hydrologic and water quality impacts along with biomass yield. The SWAT model crop database includes a large range of crops, but does not have energy crops such as miscanthus. However, researchers have developed model parameters for the miscanthus crop for SWAT application to simulate its annual biomass yield [21,34]. In this study, we used newly developed parameter sets for miscanthus crop as described in Trybula et al. [34] to predict miscanthus yield potential and its impact on stream water quality.

### 2.4. Model Setup and Inputs

An ArcGIS (ESRI$^{TM}$, Redlands, CA, USA) extension—graphical user interface of SWAT—ArcSWAT (ver. 2012) was used to build a watershed model using inputs such as topography, soil, climate, vegetation, and land management practices. A DEM (30 m) from the USDA-NRCS Geospatial Data Gateway (https://datagateway.nrcs.usda.gov/) was used to delineate the watershed. The Gridded SSURGO (gSSURGO) and CDL (in the year 2014 with identified SML) were used as soil and land cover data, respectively. The watershed was subdivided into 396 sub-basins with a threshold area of 400 ha. The watershed was further subdivided into 1464 HRUs on the basis of land cover, soil type, and slope. A threshold value of 10%, 20%, and 10% was used for land use, soil, and slope, respectively, in the HRU definition to avoid smaller HRUs. The land use for corn, soybeans, and SML was very small as compared to that of pasture and forest lands. During the HRU definition execution process, we excluded land use for corn, soybean, and strip-mined land to avoid merging small land use classes such as corn, soybeans, and SML.

Climate data from 2000 to 2013 including daily precipitation, temperature, solar radiation, relative humidity, and wind speed were collected from Texas A&M University for SWAT (https://globalweather.tamu.edu/) [35]. Daily stream discharge for three outlets in the study area was obtained from the USGS website (https://waterdata.usgs.gov/nwis/rt) for three gauges (USGS 03111500, 03111548, and 03113990). Sediment and nutrient data were collected from Ohio-EPA for the study area (Figure 2). Water quality data were available for only one USGS gauge (i.e., USGS 03111548) and were collected by the United States Environmental Protection Agency (U.S. EPA, C03S18) during the study period. Data from 2008 to 2011, and 2012 to 2013, were used for model calibration and validation, respectively. Scenario analysis was performed from 2004 to 2013 to estimate miscanthus yield in strip-mined land and its effect on flow and water quality. The average annual rate of atmospheric deposition for ammonium and nitrate in precipitation (0.24 and 0.74 mg L$^{-1}$, respectively) and dry deposition (0.59 kg NH$_4^+$ ha$^{-1}$ and 0.17 kg NO$_3^-$ ha$^{-1}$) was obtained from Clean Air Status and Trends Network (CASTNET) data for Ohio [36].

### 2.5. Crop Management Inputs to the SWAT Model

Crop management practices such as crop rotations, tillage, and fertilizer input rates play a significant role in plant growth, and impact stream water quality. It is difficult to collect management information for all land use and crop types in a large watershed. Forest and pasture land cover types covered 90% of the area in the watershed. Corn and soybeans are the two major crops cultivated in the study area. A two-year crop rotation was considered for corn and soybean croplands with local management inputs as mentioned in Cibin et al. [37] but considering no removal of corn stover. We built the crop rotations including all required field operational dates for each crop and uploaded into SWAT as management inputs for all land areas with corn and soybeans in the study area (Table S3, supporting document).

Miscanthus was represented in the model as multi-year crop rotations with the one-year establishment and 14 years of biomass harvesting (Table S3). The detailed description of the dates and management inputs are reported in Cibin et al. [13]. The soil in the SML area has low nutrient content as compared to that of cropland and it may require higher rates of nitrogen for normal crop growth. Therefore, higher rates of nitrogen (i.e., urea) fertilization were considered in this study. We modeled three scenarios: Scenario 1 (the control): no nitrogen application; Scenario 2:100 kg N ha$^{-1}$; and Scenario 3:200 kg N ha$^{-1}$).

### 2.6. Model Evaluation

The model's capability to simulate watershed responses accurately and consistently is essential to extend its use of the model. Therefore, the developed watershed model was evaluated using parameter sensitivity analysis, model calibration followed by model validation. Sensitivity analysis identifies the

most influential parameters affecting the model outputs such as flow, sediments and nutrients and their ranking [38]. It identifies sensitive parameters from a large number of model parameters and adjusts their initial ranges for further model calibration, thus helps in reducing efforts to calibrate and validate a watershed model.

The SUFI2 (Sequential Uncertainty Fitting-2) procedure in SWAT-CUP (SWAT Calibration and Uncertainty Procedures) was used for sensitivity analysis, uncertainty analysis, autocalibration, and validation of the watershed model [33,39]. The model performances were statistically evaluated using the Nash-Sutcliffe (NS) efficiency, the coefficient of determination ($R^2$) and percent of bias (PBIAS). Usually, to be considered satisfactory, monthly-simulated SWAT model water flow at the watershed scale should have an NS value > 0.55, $R^2$ values > 0.7 and PBIAS < 15% [40]. However, these values differ for sediment and nutrients (for sediment: NS value > 0.45, $R^2$ value > 0.4 and PBIAS < 20%; for Nitrogen: NS > 0.35, $R^2$ > 0.3 and PBIAS < 30%; for Phosphorous: NS > 0.4, $R^2$ > 0.4 and PBIAS < 30%) [40]. The values of NS, $R^2$, and PBIAS for flow, sediment, and nutrient results of the SWAT model simulated daily basis are typically smaller compared with yearly or monthly time-scale simulations and they depend on specific project type and difficult to get satisfactory results due to unavailability of sufficient observed data for calibration and validation [41]. In this study, the model was simulated on a daily time-step from 2000–2013 with a warm-up period of three years. The model was calibrated and validated from 2008–2011 and 2012–2013, respectively.

The initial streamflow calibration process in SWAT-CUP had 36 hydrologic parameters (Table S4) and the model was run for three sets of 1000 simulations each. After the model was calibrated for hydrology parameters, the narrow ranges for these parameter values were fixed and the model was calibrated for $NO_3$–N load using the 12 N-parameters listed in Table S4. A similar approach was used to calibrate sediment using 14 parameters affecting sediment load in the stream water. Similar to flow calibration, we performed three sets of simulations each with 1000 runs to calibrate nitrate and sediment using SWAT-CUP. The most sensitive parameters affecting nitrate and sediment are presented in Table S5.

## 3. Results and Discussion

### 3.1. Model Calibration and Validation

Streamflow was calibrated and validated on daily as well as on a monthly basis at three outlets within the watershed. Sediment and nutrients were calibrated and validated only at one outlet (due to non-availability of observed water quality data in the other two outlets) (Figure 2).

Model calibration and validation performances were evaluated based on the estimated values of the coefficient of determination ($R^2$), Nash–Sutcliffe efficiency (NS) and percent of bias (PBIAS). The model calibration and validation performance were satisfactory for monthly streamflow at all three outlets (Table 1). The calibration/validation results ($R^2$ and NS values) of flow on a daily basis is lower than monthly basis. This was due to the terrain and geography of the study area. A substantial part of the total study area is SMLs (~8%) and more lands are mined and disturbed every year—a continuous increase in the areas of SMLs in the study area. This might contribute the difficulty to achieve higher $R^2$ and NS values (for a longer duration study period) for the calibration and validation of the SWAT model. For the years in which observed nutrient data are available (2010–2011), the $R^2$ and NS values for calibration were estimated to be 0.62 and 0.59, respectively (station 03111548/C03S18), for the flow on daily basis. The calibration and validation performances for sediment and nitrate were above satisfactory limits [40]. The phosphorous concentration in the stream water at the outlet was very low (less than 0.01 mg $L^{-1}$) and was not considered for the model's calibration and validation. However, the model parameters that affect phosphorous were included during model calibration and validation of flow and sediment.

**Table 1.** Summary of calibration and validation of watershed for flow and nutrients.

| USGS/EPA Station ID | Output | Calibration (2008−2011) | | | | | | Validation (2012−2013) | | | | | |
|---|---|---|---|---|---|---|---|---|---|---|---|---|---|
| | | Monthly | | | Daily | | | Monthly | | | Daily | | |
| | | NS | $R^2$ | PBIAS | NS | $R^2$ | PBIAS | NS | $R^2$ | PBIAS | NS | $R^2$ | PBIAS |
| 03111500 | | 0.77 | 0.79 | −8.6 | 0.46 | 0.51 | −25.2 | 0.61 | 0.70 | 12.0 | 0.45 | 0.46 | −6.1 |
| 03111548/C03S18 | Flow | 0.72 | 0.72 | −2.3 | 0.43 | 0.48 | −19.7 | 0.75 | 0.79 | −10.2 | 0.42 | 0.44 | −14.8 |
| 03113990 | | 0.67 | 0.67 | −1.2 | 0.44 | 0.49 | −13.9 | 0.69 | 0.77 | 17.3 | 0.41 | 0.43 | 13.1 |
| 03111548/C03S18 | Sediment | | | | 0.86 | 0.87 | −8.3 | | | | 0.90 | 0.93 | 9.5 |
| 03111548/C03S18 | Nitrate (NO$_3$-N) | | | | 0.91 | 0.93 | 9.2 | | | | 0.70 | 0.73 | 12.1 |

Figures 3–5 present the calibrated and validated hydrograph, sediment discharge, and nitrate (NO$_3$-N) load in the stream outlet, respectively. The hydrograph illustrates that the model adequately predicts the low flow as well as peaks (Figure 3). Although the model performance was satisfactory, the fluctuations in sediment discharge during calibration and validation periods were large especially during the validation period (Figures 4 and 5). A large percentage of total area in the drainage area is SML, which could have contributed to a large fluctuation in the sediment and nitrate loads in the stream water. However, all sensitive model parameters during calibration were used for analyzing results. Furthermore, the number of observed data points is very small, and the model could be further improved with additional observed data points related to water quality for calibration and validation.

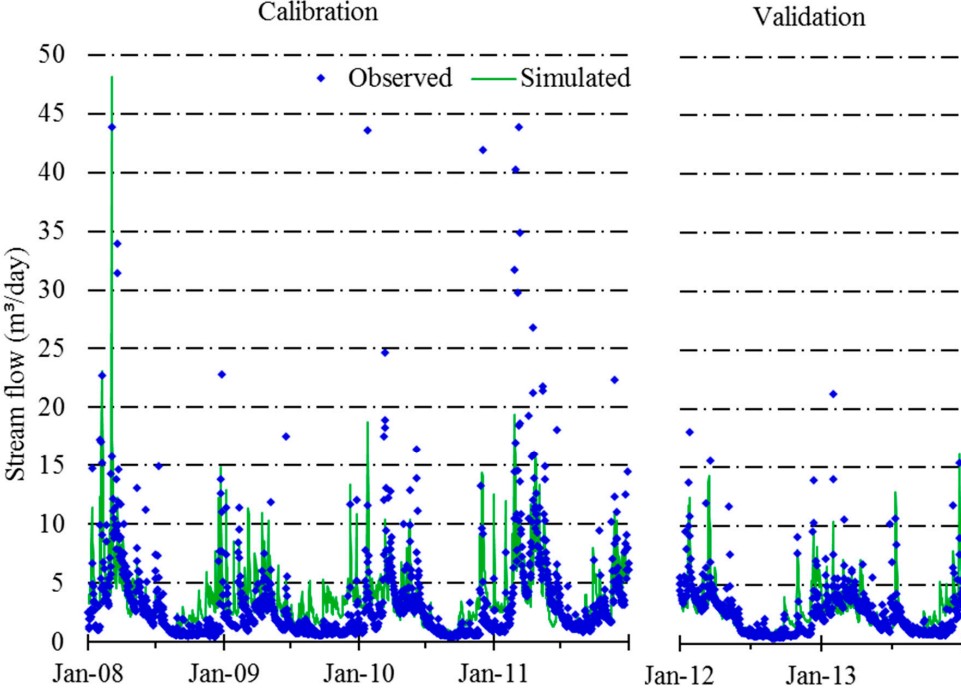

**Figure 3.** Hydrograph during calibration (2008–2011) and validation (2012–2013) periods.

With the limited number of available observed data points, the model was able to predict nitrate loads during wet and dry seasons (Figure 5). The trends of fluctuation of nitrate load during calibration and validation period were similar to flow. In overall, the performance of calibration and validation were satisfactory and further used in estimating the effect of growing energy crops in the SML on watershed performances. The best parameter values in the calibration are used for simulating the watershed to estimate flow, sediment, nitrate, phosphorous, and biomass yield. The results from the models were analyzed and aggregated to watershed and sub-basin level for further presentation and analysis.

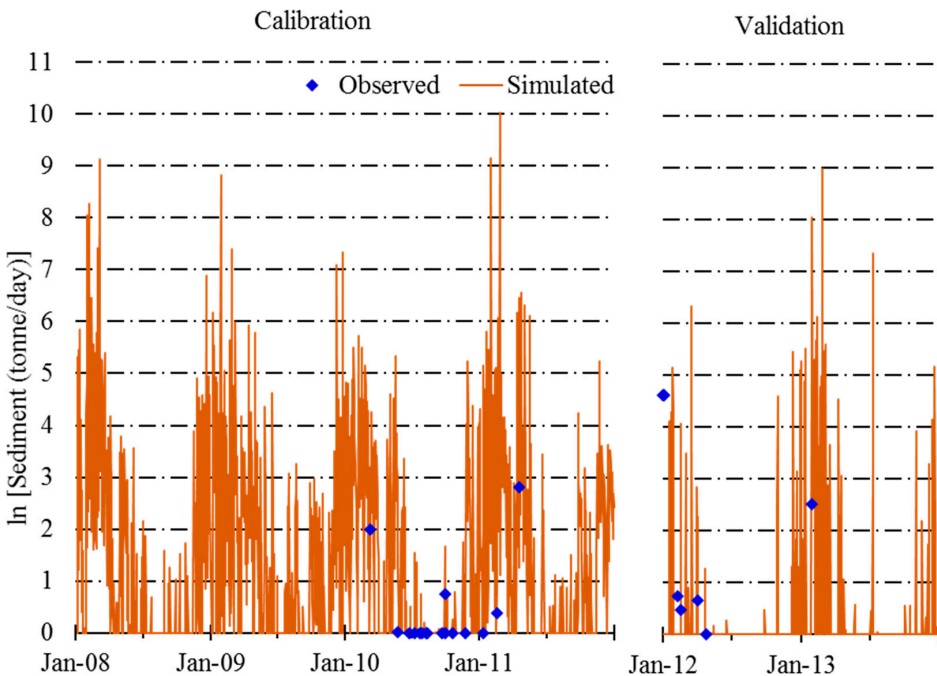

**Figure 4.** Calibration and validation of sediment loading to the stream.

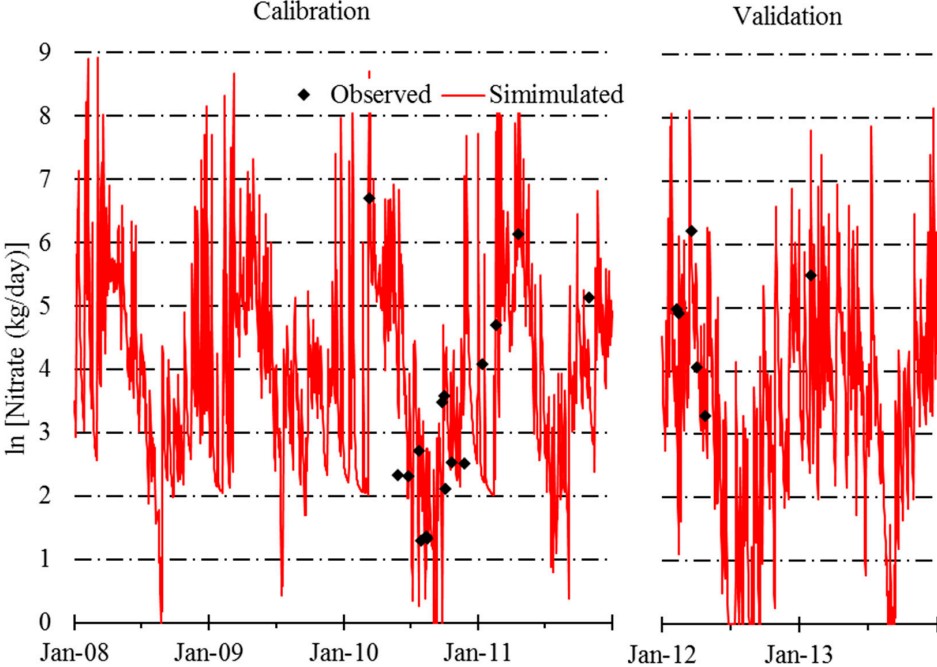

**Figure 5.** Calibration and validation of nitrate (NO$_3$-N) loading to the stream.

### 3.2. Miscanthus Cultivation in SML Impacting Watershed

The performance parameters to evaluate the option to grow miscanthus in the SML for biomass production were analyzed with the help of average annual flow and sediment, nitrate, and phosphorus loading at the outlets (Figure 6, base case vs. scenario 1 only) as well as at the sub-basin level (Figure 7, comparative results for all scenarios).

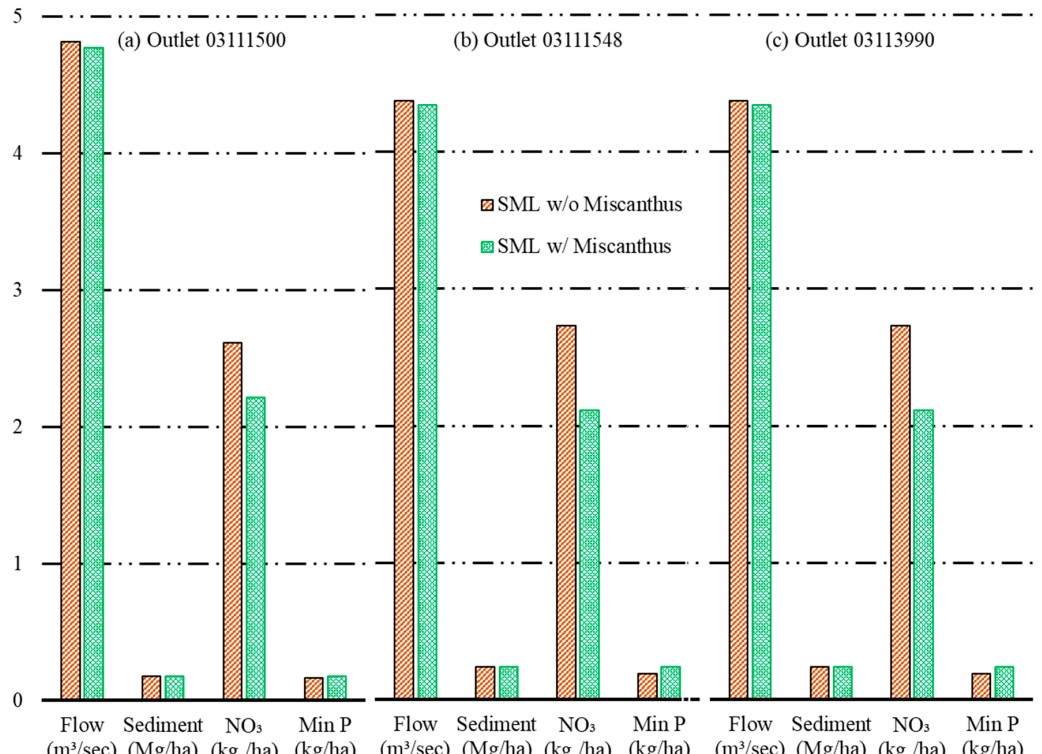

**Figure 6.** Variations in simulated average annual flow and water quality parameters values at the stream outlets due to growing miscanthus in SMLs (scenario 1) in the studied watershed. (**a**) Outlet 03111500; (**b**) Outlet 03111548; (**c**) Outlet 03113990.

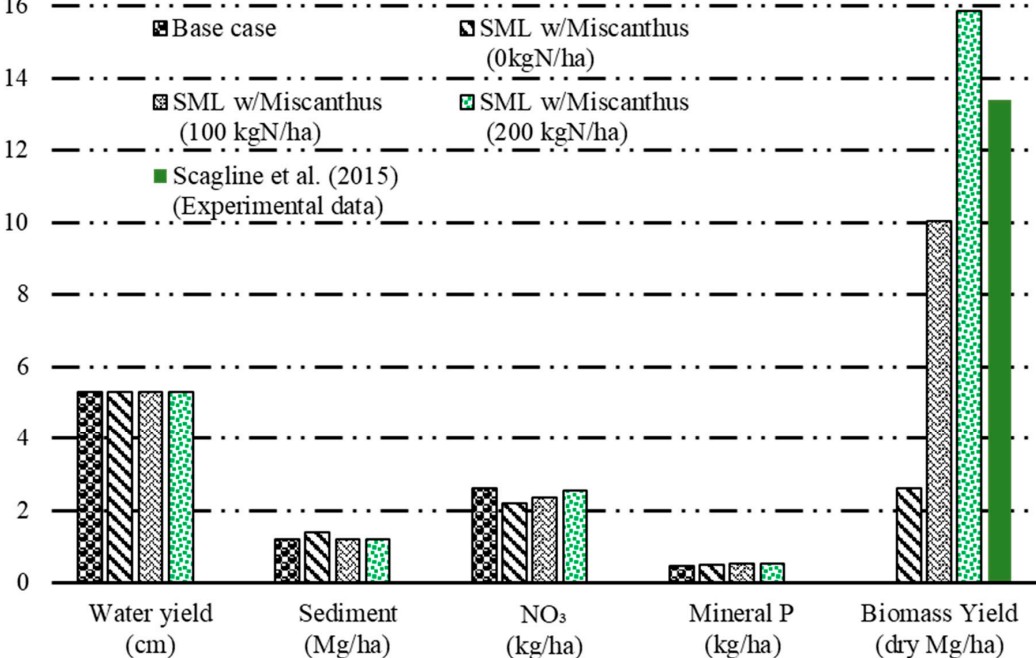

**Figure 7.** Simulated annual average water yield, nutrients load and biomass yield at the basin level considering the different level of fertilizer application.

Streamflow and sediment loading at the watershed outlets were slightly reduced under miscanthus production scenarios in the SML compared with the base case. Compared with the base case, the stream water flows at three watershed outlets were reduced by ~1% in scenario 1

(Figure 6). The reduction in the nitrate loading (7.7–22.5%) to the stream outlets was higher compared with the flow and sediment.

Miscanthus is a perennial crop and its establishment on SML reduced the surface flow by providing physical barriers, and increased infiltrations and evaporation that caused lesser sediment load to the streamflow. The net nitrate load reduction from drainage areas at the outlets-b and outlets-c were larger than that of outlets-a due to the presence of a larger proportion of SML in the drainage areas for the former two outlets compared with the later (i.e., outlet-a, Figure 6). The mineral phosphorous load at the outlets was increased due to the mineral phosphorus application during miscanthus cultivation. Nitrate leaching to stream water led to nitrous oxide emissions (highly potent greenhouse gas emissions) into the atmosphere. Thus, the reduction in nitrate loads to stream water can reduce the net GHG emissions from fertilizer applications (or lower carbon footprint of biofuel or bioenergy production from biomass produced in SMLs) as well as reduce eutrophication potentials in rivers and streams [42].

Figure 7 shows the basin-level weighted annual average streamflow and water quality parameters by growing energy crops in the SML with different fertilization levels (scenarios 1 to 3) and base case (no energy crops). The overall changes in the annual average water yield (in cm $H_2O$) were marginal (<1%) at the watershed level and moderate (1–6%) at the basin and sub-basin level, respectively, in all miscanthus-growing scenarios in SMLs.

Scenario 1 showed a substantial increase in basin sediment yield, i.e., by 27.0 (very large at sub-basin level, −10 to 800% not presented here) with respect to the base case. The large sediment yield increase was due to very low miscanthus yield and biomass removal (e.g., ~70.0%) or leaving a little residue on the SML to obstruct surface flow from precipitation.

A large portion of the SML falls within higher slope categories and soil erosion increased exponentially with the removal of biomass from these lands [43]. However, with an increase in the biomass yield (i.e., scenarios 2 and 3), the sediment load decreased. Higher biomass yield in scenario 3 reduced the flow as well as soil erosion and the average annual sediment yield was lower than that of the base case.

The nitrate load trend was increased with increasing fertilization rate during miscanthus cultivation on SML (Figure 7). In scenario 1, nitrate load decreased substantially (i.e., by 14.0%) compared with that of the base case. Then nitrate load to the stream water from sub-basins increased with the addition of more nitrogen fertilizer during miscanthus cultivation. Although higher nitrogen application rates for miscanthus cultivation increased the nitrate loads in the stream water, still these values at higher nitrogen fertilizers application rates were less than the base case scenario (no miscanthus in SML). The net reduction of nitrate load at the basin level was 9.0% (scenario 2, 100 kg N $ha^{-1}$) and 2.5% (scenario 3, 200 kg N $ha^{-1}$), respectively. Most of the nitrogen applied during miscanthus cultivation was absorbed by the plant and induced growth.

The net increase in mineral phosphorous was ranged from 10 to 12% due to either higher sediment load (scenario 1, phosphorous is attached to the sediment) or the additional phosphorous used during miscanthus cultivation (75 kg $ha^{-1}$ in scenarios 1 and 2).

The increase in nitrogen fertilization rate during miscanthus cultivation increased the biomass yield by about 4-fold (scenario 2, 100 kg N $ha^{-1}$) and 6-fold (scenario 3, 200 kg N $ha^{-1}$) compared to the zero fertilization rate (Scenario 1). The various nitrogen application rates to cultivate miscanthus simulate the biomass yield potential [44,45]. However, a higher rate of fertilizer application could increase the $N_2O$ (a potent GHG emission agent) emission and nitrate leaching (a eutrophication agent) that may reduce the environmental benefits. However, higher biomass yield has a substantial effect on reducing feedstocks cost for biofuel production and improve its economic feasibility [19,46,47]. Thus, an integrated techno-economic study along with the SWAT model is necessary to break-even the economic and environmental advantage of growing miscanthus in SML.

Figure 8 shows the relative change in the watershed performance at the sub-basin level by different miscanthus cultivation scenarios with respect to that of the baseline (a1 − a5 = $\frac{scenario\ 1}{base\ case}$, b1 − b5 =

$\frac{scenario\ 2}{base\ case}$, and c1 − c5 = $\frac{scenario\ 3}{base\ case}$). Figure 8 (a5, b5, and c5) shows the spatial average annual biomass yield at three fertilizer application rates. A marginal decrease in the water yield (i.e., surface flow) was experienced by most sub-basins. The spatial distributions of the sub-basin level changes in the sediment yield, nitrate and phosphorous yield are similar throughout the watershed. Figure 8 also illustrated that the variations in the performance indicators of scenarios 1–3 (i.e., growing miscanthus on SMLs) compared with the base case were wide among sub-basins in the watershed. Thus, each sub-basin requires unique management practices that can provide optimal benefits to the watershed. For example, although fertilizer application rate was the same for the entire watershed, biomass yield in sub-basins towards the center of the watershed were very low compared with that of other sub-basins that might unnecessarily increase the nitrate load to the stream water. Therefore, SML in sub-basins with very low miscanthus yield could be cultivated with no or little fertilizer or remain unchanged as before to achieve the overall sustainability objectives.

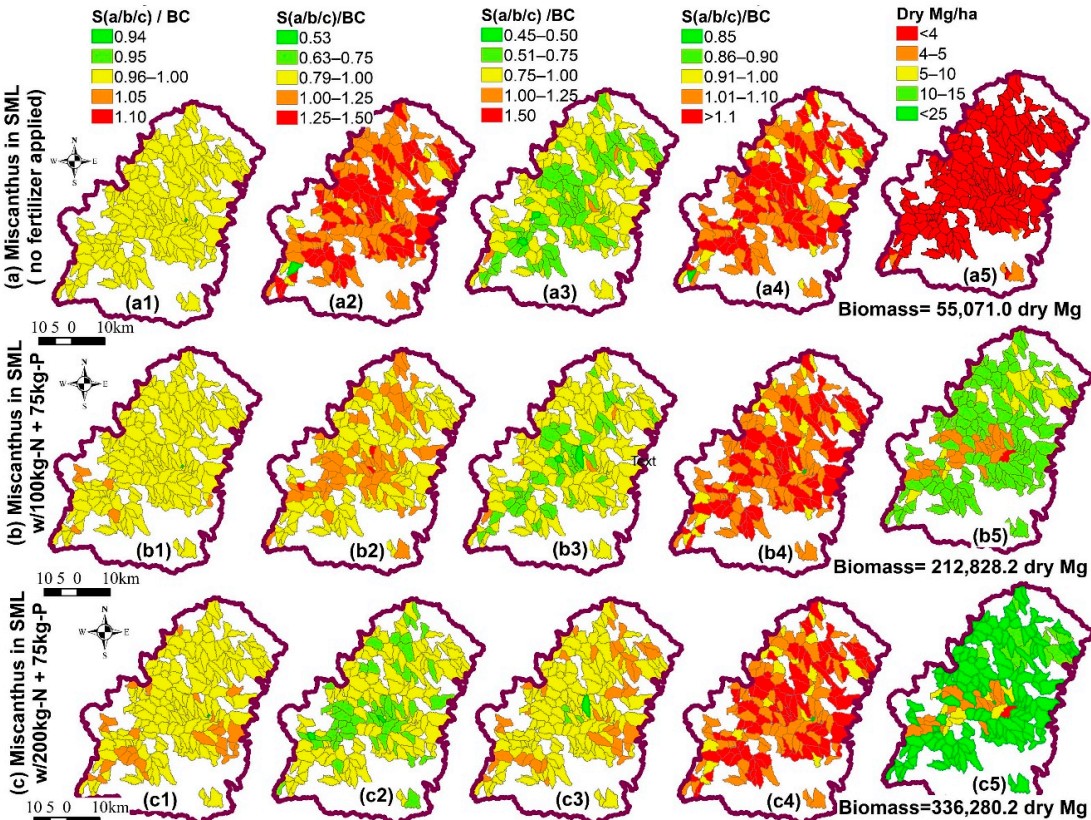

**Figure 8.** Impact of growing energy crops (i.e., miscanthus) in SML on watershed performances and potential of annual biomass availability considering different scenarios (no (**a**)/low (**b**)/high (**c**)) of nitrogen application (BC: Base case; **1**: Water yield, **2**: Sediment, **3**: Nitrate, **4**: Phosphorous, and **5**: Biomass yield).

### 3.3. Miscanthus Biomass Availability

The climate, soil, and management practices had a substantial impact on biomass yield and their variations are not uniform throughout the watershed. Figures 7 and 8 (c5) provide the annual average biomass yield during the study period and spatial variations in the biomass yield for the studied area, a watershed in the state of Ohio. The temporal variations of biomass yield are shown in Figure 9. The annual variations in the biomass yield can be attributed to the physiology of energy crops, and climate factors such as rainfall and temperature.

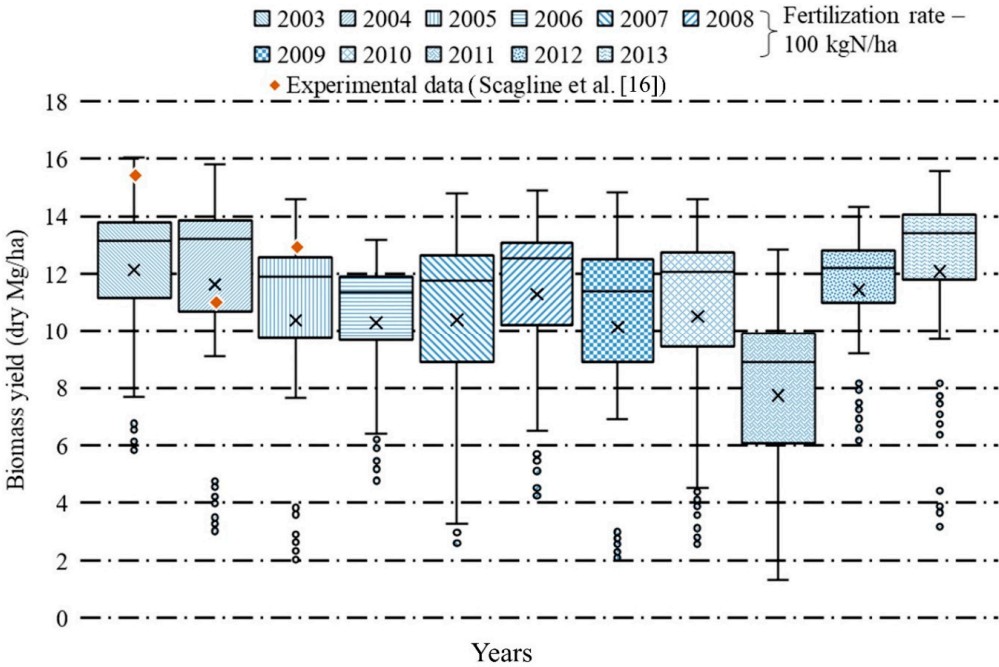

**Figure 9.** Temporal and spatial variations in annual biomass yield of miscanthus produced from SML (100 kg N ha$^{-1}$ and 75 kg P ha$^{-1}$).

The yield predicted in this study with application of nitrogen was very similar to experimental biomass yield results from SML [15–17]. The average miscanthus yields in SMLs without the application of fertilizer (scenario 1), with the application of 100 kg N and 75 kg P (scenario 2), and with application of 200 kg N and 75 kg P (scenario 3) were 2.6 (0.8–5.53), 10.4 (1.32–16.05), and 15.86 (1.34–26.03) dry Mg ha$^{-1}$, respectively. However, the increase in the biomass yield in this study (~600% increase between scenario 3 and base case) was much larger than in experimental studies [44,48]. Arundale et al. [48] quantified a 25% increase in the biomass yield in miscanthus cultivated in cropland with 202 kgN/ha compared with no N fertilizer application [48]. Compared with cropland, SML has very low-quality soil and deficient in nutrient especially nitrogen that hinders plant growth [22]. Therefore, a large increase in the biomass yield with a higher rate of nitrogen application was observed in this study. However, the rate of increase in biomass yield was lower than the rate of increase in nitrogen fertilizer application and the trends are similar for both cropland and SML. Other than soil quality, biomass yield also depended on climate parameters such as rainfall, temperature, sunlight, etc.

The study illustrated that SML requires fertilizer to achieve higher yield but the increase in the biomass yield with respect to increasing fertilizer application rate is not linear, and a higher fertilizer application rate has a detrimental effect on stream water quality. However, higher biomass yield could lower the biomass logistics cost and improve the economic feasibility of making biofuel or bioenergy from biomass. Nitrogen application at a high rate could also have a negative impact on overall GHG emissions. Therefore, a more in-depth and holistic experimental study is required to find the optimal fertilizer application rate for miscanthus considering both environmental and economic factors.

In the current study area (21,203 ha of SML), the annual production of miscanthus varies between 55,000 and 336,000 dry Mg. Post-SMCRA law, there are about one Mha (Table S1) and 73,000 ha (Table S2) of SML available in the USA and Ohio, respectively [11]. These lands could supply about (10–15) and (0.7–1.0) M dry Mg of biomass every year to produce bioenergy. Although predicted miscanthus biomass yields from SMLs are much lower than that of usual croplands with fertile soil [8], the biomass yields are still much higher than that of other perennial energy crops such as switchgrass. Therefore, the production of miscanthus from SMLs could be economical and it can provide a substantial amount of biomass for bioenergy or biofuel production without competing for land to produce food and feed for the growing human population. Furthermore, the production

of biomass from these lands also has positive environmental benefits such as reduction in stream sediment yield (~20%) and nitrate load (2.5–10.0%), with only a marginal increase in the mineral phosphorous yield to the stream water.

Currently, major stakeholders such as mining contractors must reclaim the SMLs after mining activities, which is a very expensive option for them [31]. Reclamation of disturbed SMLs is a lengthy process (i.e., 5–10 years)—growing plants is one of the most critical processes during the reclamation that adds soil organic carbon to the disturbed l and—and restores the soil productivity similar to conditions before mining activity. Miscanthus is also one of the best plant species to reclaim SMLs because it is better adapted to grow in nutrient- and water-deficient soils, and offers faster carbon accumulation—that is, sequesters more carbon quickly. Therefore, growing miscanthus in the SML can provide biomass for energy production, and environmental benefits while reducing the cost of reclamation, especially for mining contractors.

Although the SWAT model was calibrated and validated with observed values of stream flow, sediment, and nutrients, the number of observed values were limited in quantity during the study period. The results of the study should encourage the scientific community to collect and generate more data related to watershed performances and miscanthus biomass yield. This will help in reducing the uncertainties and improve the quality and reliability of the results. This is the first step towards recognizing the potential multiple benefits of SML use for biofuel production and strengthening the bioeconomy, as well as making the transition to integrated solutions to achieve land degradation neutrality [49] in order to make progress towards United Nations' multiple Sustainable Development Goals.

## 4. Conclusions

A GIS-based hydrological watershed model was developed to estimate the miscanthus yield potential, while assessing the soil and water quality on the strip-mined lands (SMLs) using the SWAT model. This study has demonstrated that establishing miscanthus in the SML could produce about 2.6, 10.0, and 16.0 dry Mg of biomass assuming fertilizer application rates of 0, 100, and 200 kg N ha$^{-1}$, respectively. The environmental benefits of growing miscanthus potentially reduced the loading of sediments (~20%) and nitrates ($NO_3$–N, 2.5–10.0%) in the stream water compared to that of the base case (without energy crop cultivation in SML). The available SML in the USA could produce about 10.0–16.0 M dry Mg year$^{-1}$ of biomass sustainably for bioenergy production. This study has also identified hotspots (i.e., most influential SML areas) in the study region for sustainable biomass production. The predicted biomass yield from SML was comparable with experimental data and the cultivation cost can be economical with positive environmental benefits. The developed SWAT model for the strip-mined lands can be used by mining contractors, policymakers and bioenergy investors to assess the economic and environmental benefits of producing biomass for bioenergy applications. In the future, the economic impacts of growing miscanthus to produce biofuels on SMLs need to be evaluated, if the biomass cultivation may be integrated with the existing soil reclamation practices in SMLs.

**Supplementary Materials:** The followings are available online at http://www.mdpi.com/2073-4441/11/3/546/s1, Table S1: Strip-mined lands stocks in the US from 1977 to 2012, Table S2: Strip-mined lands stocks in the state of Ohio from 1977 to 2012, Table S3: SWAT management inputs for annual and perennial crops, Table S4: Stream-flow, nitrogen and sediment parameters and minimum, and maximum values used for calibration period using the SWAT-CUP, Table S5: SWAT input parameters included in the calibration process and their calibrated values and sensitivity statistics.

**Author Contributions:** K.S., S.M., and A.M.M. came with the research idea; K.S. and A.M.M. designed the project and modeling approach; K.S. performed, and N.H., A.M.M., and S.S.P. assisted with, the model setup, simulation, and calibration; K.S., S.M., A.M.M., N.H., and S.S.P. analyzed the results and developed the manuscript's structure; all authors contributed to writing and revising the paper; and project administration and funding acquisition for the study was done by S.M.

**Funding:** This project was partly supported by the USDA-NIFA Biomass Research and Development Initiative (BRDI) grant (Grant # 2012-1008-2032).

**Acknowledgments:** The authors would like to thank Donovan Powers, ODNR (Ohio Department of Natural Resources) for proving GIS maps of strip-mined lands for the state of Ohio. Special appreciations to David E. Radcliffe at the University of Georgia for his guidance on parameter selection in the SWAT model for nutrient calibration and validations and analyzing water quality results.

**Conflicts of Interest:** The authors declare no conflict of interest.

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
