# Peer review of "Assessment of Miscanthus Yield Potential from Strip-Mined Lands (SML) and Its Impacts on Stream Water Quality"

_water, doi:10.3390/w11030546_

Round 1

Reviewer 1 Report

Great work. I find no significant flaws although using SWAT for simulating crop yields is outside of my area of expertise. The modeling presented seems fine and I look forward to seeing this work in print.

Author Response

We appreciate reviewers for their valuable comments on this manuscript. The responses to those comments were accommodated in the revised version of the manuscript.

Reviewer #1:

Overall Comment: Great work. I find no significant flaws although using SWAT for simulating crop yields is outside of my area of expertise. The modeling presented seems fine and I look forward to seeing this work in print.

Response: Thank you. We appreciate your comments.

Reviewer 2 Report

Good manuscript on an important topic. As commented in the manuscript, I find some over-extrapolated and under-supported statements in the Introduction related to food-vs-fuel, ILUC, and biofuels policy (lines 38-47). I don't think there's a need to claim that the potential food-vs-fuel and ILUC issues justify investigation of energy crop production on SMLs. See my comments tracked in the manuscript.

Author Response

We appreciate reviewers for their valuable comments on this manuscript. The responses to those comments were accommodated in the revised version of the manuscript.

Reviewer #2:

Overall Comment: Good manuscript on an important topic. As commented in the manuscript, I find some over-extrapolated and under-supported statements in the Introduction related to food-vs-fuel, ILUC, and biofuels policy (lines 38-47). I don't think there's a need to claim that the potential food-vs-fuel and ILUC issues justify investigation of energy crop production on SMLs. See my comments tracked in the manuscript.

Response: Thank you. We revised the manuscript based on your inputs.

Comment 1:  ...is "a" rather than "the" non-cropland

Response: Revised

Comment 2. I don't believe reference #7 concludes categorically that miscanthus and switchgrass are "...proven as the most suitable non-invasive plant species to produce biomass...". In fact, reference #7 only evaluate these two energy crops. "Most suitable to produce biomass" would depend on any scenario-specific objectives. Please refine this statement to be more objective.

Response: Revised

Comment 3. I realize citation #8 says this, but "large cultivated area" compared to what"? What if 200 million of the estimated 300 million tons comes from residues? Please change to a specific area under specified assumptions and put within context agricultural land in the US to be more specific.

Response: Revised. About 15.8 Mha of land required to produce 60 Giga liters of cellulosic biofuel (RFS mandate by 2022), which is about 10% of total agricultural lands in the U.S. (157.8 Mha in 2012).

Comment 4. Just because concerns of food insecurity and ILUC are raised for some doesn't mean that biomass production has to be limited to non-ag land as might be suggested in this statement. If we really needed to be more efficient with food production from land then we shouldn't allocate 1/3 of corn and soy production to animal feed. Please refine this sentence to be more specific and objective.

Response: Thank you. We agree completely with the reviewer. Therefore, we reduced the tone of this sentence. We just cited previously published research and identify there may be a concern from a certain section of the researchers. Our intention was to provide the reader with information that there is another type of lands (such as marginal lands, CRP lands, and strip-mined lands) available to grow biomass for biofuel production.

Comment 5. They are low in nutrients... if they are targeted...? So if they are not targeted they are not low in nutrients? This doesn't make sense.

Response: Revised. This sentence was removed to reduce the confusion.

Comment 6. "huge" is a relative term (and kinda slangy).

Response: Revised.

Comment 7. Maybe I'm missing it, I'm not good with statistics. But is there enough correlation in Figure 4 to do what you want with this relationship?

Response: Yes. Details statistical results are provided in Table 1.

Comment 8. "water yield" is defined below, but would be better defined at first use on the term.

Response: Revised.

Comment 9. Consider comparing with and referencing https://onlinelibrary.wiley.com/doi/full/10.1002/bbb.342.

Response: Thank you. The article referred here was for switchgrass. Moreover, this study is specific to only Strip-mined land.  Therefore, we could not compare the results from the article with this study.

Comment 10. Are (a) (b) and (c) the same as the scenarios? If so, make consistent 1/2/3 a/b/c?.

Response: Thank you. These are scenarios. Changing from a/b/c to 1/2/3 will create a problem with subsections, i.e., a1-a5, b1-b5, and c1-c5.

Comment 11. "Relative change" rather than impact?

Response: Yes.

Comment 12. as defined in the equations in the text? Add to the figure caption so the figure stands alone.

Response: Revised

Reviewer 3 Report

Please see the attached review report.

Author Response

We appreciate reviewers for their valuable comments on this manuscript. The responses to those comments were accommodated in the revised version of the manuscript.

Reviewer #3:

Overall Comment: The objective of this manuscript is to simulate miscanthus biomass yield and availability of miscanthus from Striped-Mined Lands in a watershed in Ohio using the SWAT model. This study is in the scope of Water. This paper is well organized and well written generally. The importance of this research was well described. The scientific results and conclusions were not presented in a clear, concise, and well-structured way. Generally, this study is innovative and comprehensive. More in-depth discussion should be included to support the interpretations and conclusions.

Response: Thank you. We appreciate your comments. The results and conclusions sections are revised based on your comments.

Comment 1. What is the novel idea this manuscript provided to scientific knowledge? Please describe it and use your results and discussion to support it.

Response: Thank you. Revised.

In the U.S., a substantial agricultural land area (about 3.5 Mha lands) had been disturbed by mining activity only after 1977 and this number will be higher if we consider disturbed lands before 1977.  Growing energy crops in strip-mined lands can accelerate the reclamation process as well as improve stream water quality by preventing soil erosion and other nutrients and chemicals. Finally, this can be a win-win situation for everyone, i.e., mine operators, biofuel industry, and local society. There are only a few experimental studies have tried to grow energy crops and found positive results. However, there is no study to analyze the impacts of growing energy crops on large scale in SML. This is the first attempt to simulate and analyze the impacts of growing energy crops, such as miscanthus on strip-mined lands. Our study has demonstrated that the impacts of growing energy crops in SML on water quality as well as predicted the potential yield with different management practices. The article provides new and novel information to the scientific community and different stakeholders to improve strip-mined lands in the U.S.

Comment 2. The conclusions did not interpolate researching findings well. What are boarder impacts of these results? How will it be beneficial for watershed management? It would be more meaningful if authors can incorporate this information.

Response: Thank you. We had incorporated this information in the conclusion section.

The conclusion section shows the reduction of sediment and nitrate in the stream water in the watershed having SML and cultivated with energy crop, such as miscanthus. It also shows the predicted biomass yield with different management practices, i.e., the use of nitrogen fertilizer. Moreover, the potential of producing energy crop from the study area as well as all SML in the U.S. has been mentioned.

Comment 3. What is limitation of this study? What about the uncertainty in the model inputs? How to improve it in the further study? What kind of take-home messages you would like to deliver to readers? Any ideas about how to disseminate the research results to the potential stakeholders? Have the authors received any feedback to this study from the potential stakeholders yet?

Response: Although the SWAT model was calibrated and validated with observed values of stream flow, sediment, and nutrients, the observed values were limited in terms of numbers and study period. The results of the study may encourage the scientific community to collect more data related to watershed performances and miscanthus biomass yield. This will help in reducing the uncertainties and improve the quality and reliability of the results. This is the first step towards recognizing the multiple potential benefits of SML use for the biofuel production and strengthen bioeconomy. 

The developed simulation model provides a platform to assess and improve the sustainable management of mined lands during reclamation and to generate additional revenue through selling as a bioenergy feedstock by the potential stakeholders (i.e., mine operators, bioenergy producers, landowners, and local community).

In the future, we will work with NGO’s and other private organizations who are promoting energy crops cultivation in mined soils as they are released for reclamation activities to support their efforts. Our results demonstrated that the environmental impacts due to water quality is minimal with existing watershed data. We hope that further expansion of mined soil reclamation activities will provide additional data to further validate and reassess the impacts of water quality and biomass yield potential for bioenergy applications.

Comment 4. Line 87: Please explain “a large study area (the state of Ohio)”. It sounds like this study was conducted in the entire Ohio state. However, the model was calibrated, validated and simulated in a small watershed in Ohio instead of the entire Ohio. Please rewording to avoid confusion.

Response: The methodology section shows the study area and related information. The large study area in the state of Ohio, means the size of the study area (2632.5 km2). Figure 2 shows both the locations of SML in the state of Ohio and the study area for which SWAT model was developed, calibrated and validated. Limiting our study to only three counties in Ohio was due to data limitations, i.e., observed data for sediment and nutrient was available only for these three counties. Other counties which contain SML did not have data to calibrate and validate the model.

Comment 5. Section 2.1.1 can be condensed and moved to the Introduction part.

Response: This section is necessary and useful to the readers, which provide information about the availability of SML and its characteristics. This section also provides new information to the readers which are not available in the public domain in a compiled manner.

Comment 6. Section 2.3 is the basic description of the SWAT model, which has been mentioned in a large number of previous studies. This section can be condensed and moved to the Introduction part. I believe this will improve the flow of the manuscript and the readers can focus more on the model setup of this case study instead of the general description of SWAT model setup.

Response: We followed the general structure of manuscript related to SWAT model. This section may help readers to understand the basic working of SWAT tool, who is not an expert in this type of work.

Comment 7. Section 2.4, which version of SWAT executable file was used in this study?

Response: The revision 637 was used in this study.

Comment 8. Terms “striped-mined lands”, “Strip-mined land”, “strip-mind land” and “Striped‐Mined Lands” were used in the title, abstract and main manuscript. Please be consistent throughout the entire manuscript.

Response: Revised.

Comment 9. Please be consistent with the past tense or the current tense. For example, the paragraph from line 125 to line 136 jumps between these two tenses.

Response: Revised

Comment 10. Line 259: “The hydrograph illustrates that the model adequately predicts the base flow as well as peaks (Figure 3).”. Figure 3 is for stream flow. I do not see the separation of stream flow as base flow and surface runoff. Therefore, I guess what you want to describe is “the model adequately predicts the low flows as well as peaks”?

Response: Revised

Comment 11.

Line 43, “rise of food prices” or “the rise of food prices”?

Line 56, “Appalachian region” or “the Appalachian region”?

Line 70, “soil organic-carbon” or “soil organic carbon”?

Line 92, “landuse” or “land use”? Please also correct it in other places throughout the manuscript.

Line 102, “the environmental degradation” or “environmental degradation”?

Line 135, “merged to” or “merged with”? “the” may need to be added before “SML category”?

Line 144, “an annual average precipitation” or “annual average precipitation”?

Line 184, “the” before “Ohio-EPA” may need to be deleted.

Line 205, “the” may need to be added before “SML area”.

Line 226, “the” may need to be added before “SWAT model”.

Line 244, “the” may need to be added before “other two outlets”.

Line 283, “the” may need to be added before “stream outlets”.

Line 293, “were larger” or “was larger”?

Line 300, Is there a dot missed at the end of the sentence?

Line 302, “basin level” or “a/the basin level”?

Line 337, “the” may need to be added before “SWAT model”.

Line 349, “the” may need to be added before “base case”.

Line 351, “the” may need to be added before “same” and “entire”.

Line 368, “the” may need to be added before “application”.

Line 431, “the” may need to be added before “conceptualization”.

Response: The revision

Reviewer 4 Report

In this manuscript, miscanthus yield was investigated by using the SWAT model in strip-mined lands. In general, the manuscript is well-written and I think the given work in very interesting. However, I cannot recommend for publication until the following issues can be addressed properly:

1. It seems that authors are using HUC8 maps for SWAT delineation. On the other hand, the given strip-mind lands are a lot small than a big HUC8. Authors need to justify why you did not adopt HUC12 instead, since quite a few research work can be found in the Great Lakes Region (Western Lake Erie for example) by using even finer resolution map such as NHDPlus?

2. The given statistical results for sediment and nitrate are a lot better than flow. On the other hand, statistics for daily flow are mostly not so good in general sense (NSE<0.5). I’m not convinced the corresponding results for nutrients are reliable. There are some studies that explored uncertainty sources of the SWAT model. I think authors should spend more time on this issue.

3. The current format of the Conclusion is relatively thin. In addition, I don’t think the developed SWAT model can be used by others (contractors, decision makers) for the reason I mentioned earlier.

Author Response

We appreciate reviewers for their valuable comments on this manuscript. The responses to those comments were accommodated in the revised version of the manuscript.

Reviewer #4:

Overall Comment: In this manuscript, miscanthus yield was investigated by using the SWAT model in strip-mined lands. In general, the manuscript is well-written and I think the given work in very interesting. However, I cannot recommend for publication until the following issues can be addressed properly:

Response: Thank you. We have revised the manuscript to address your comments.

Comment 1. It seems that authors are using HUC8 maps for SWAT delineation. On the other hand, the given strip-mind lands are a lot small than a big HUC8. Authors need to justify why you did not adopt HUC12 instead, since quite a few research work can be found in the Great Lakes Region (Western Lake Erie for example) by using even finer resolution map such as NHDPlus?

Response: Thank you. We did not use HUC8 maps for SWAT delineation. We ArcSWAT tool to delineate watershed with maximum sub-basin area of only 400 ha. Instead, these sub-basins are much finer than HUC12.

Comment 2. The given statistical results for sediment and nitrate are a lot better than flow. On the other hand, statistics for daily flow are mostly not so good in general sense (NSE<0.5). I’m not convinced the corresponding results for nutrients are reliable. There are some studies that explored uncertainty sources of the SWAT model. I think authors should spend more time on this issue.

Response: Thank you. The current study area is extremely scarce in observed data, especially for nutrient and sediment. We used SWAT-CUP to calibrate and validate the model. Therefore, the results presented here may the best that we can get from this study. Moreover, NSE values for monthly and daily simulation is more than 0.7 and 0.4 respectively, for flow, sediment, and nutrients which is satisfactory as proposed by Moriasi et al. (2015). We also performed uncertainty analysis using SWAT-CUP tool and the PBIAS values were less than 15% in monthly simulation for all model output measurements. We hope that the current study will serve as a platform to seek additional experimental data to understand the soil and water quality impacts of growing energy crops in highly disturbed and low quality mined soils. In the future, if more observed data are available, the model results can be further improved with robust results for the study area.

Comment 3. The current format of the Conclusion is relatively thin. In addition, I don’t think the developed SWAT model can be used by others (contractors, decision makers) for the reason I mentioned earlier.

Response: Thank you. We discussed uncertainty and limitations to this current work. For this study area, the available observed data for nutrients are less. In the future, if more data available, we can get more reliable and robust results using this model. 

Round 2

Reviewer 4 Report

It seems that authors are taking my comment in light ways which is kind of disappointing. I don't think one can say we have data scarcity issue in the United States, unless it is somewhere in Africa or Asian, whereas data scarcity is truly a big problem to researchers for sure. I've pointed out the given work could be completely unreliable due to the funny results. Authors cannot simply blame SWAT-CUP for it, but more work must be done in exploring why hydrologic and nutrient processes did not perform well. How can we believe that a watershed project with relatively bad flow simulation results while nutrient statistics are awesome??? I suggest sending this manuscript to independent SWAT experts and see if they agree with the situation here. 

Author Response

We appreciate reviewers for their valuable comments on this manuscript. The responses to those comments were accommodated in the revised version of the manuscript.

Reviewer #4:

Comment 1. It seems that authors are taking my comment in light ways which is kind of disappointing. I don't think one can say we have data scarcity issue in the United States, unless it is somewhere in Africa or Asian, whereas data scarcity is truly a big problem to researchers for sure. I've pointed out the given work could be completely unreliable due to the funny results. Authors cannot simply blame SWAT-CUP for it, but more work must be done in exploring why hydrologic and nutrient processes did not perform well. How can we believe that a watershed project with relatively bad flow simulation results while nutrient statistics are awesome??? I suggest sending this manuscript to independent SWAT experts and see if they agree with the situation here.

Response: We feel sorry for the misunderstanding. We understand the reviewer’s feelings and respectfully convey our sincere effort here. We tried our best to improve the calibration/validation results of the SWAT model but came with the same results as before. Here again, we tried out best to explain the situations and addressed reviewers comments.

The study area is the Appalachian region and mined lands are one of the dominant land use types. It's true that there is data limitation for this study area (and study period 2001 onwards). We have checked every publicly available database, such as USGS, STORET, etc. But we did not get the required nutrients observed data for the study area. In the end, we reached out to the local US EPA representative for water quality data personally. We got the nutrient data from them, which was used for the calibration and validation of the model.

As the number of observed data points for the nutrients are much less than observed data points for flow, the calibration/validation results of nutrients are better than the calibration/validation results of flow. In case of nutrient calibration/validation, the SWAT-CUP was easily able to rearrange the SWAT-parameters to find better-simulated results close to the observed values and thus better R2 and NS values for nutrients. But for the flow as we used daily observed data for the entire study period and the model might not able to get an optimal set of SWAT parameters that produce simulated-flow values close enough to the observed-flow data and thus produce a relatively lower quality calibration/validation result. We mentioned in the manuscript that the low calibration/validation for the daily flow was due to rough terrain (high slope) and geography (about 8% of total lands is SML and every year thousands of hectares of SML are disturbed due to mining activity) − a continuous change in the land use type in the study area. We estimated the R2 and NS values for the study period for which observed nutrient data is available (2010-2012).  For the calibration year (2010-2011), the R2 and NS values were 0.62 and 0.59 respectively for daily flow, which are much better than for the longer calibration years (2003-2010).

Moreover, daily calibration/validation results for the flow were above the satisfactory limits as defined by Moriasi et al. (2015). Moreover, the monthly calibration/validation results for the flow are good/very good (>0.7) as mentioned by Moriasi et al. (2015).

Usually calibration/validation results for the flow better than a nutrient. However, it’s not always true. There are few studies [for example Feng et al. (2015), Love and Nejadhashemi (2011)], where the nutrients calibration/validation results are much better than flow. Feng et al. (2015) had a similar situation as this case. The calibration/validation R2 and NS values for the flow (R2 =0.55-0.56 and NS=0.53-0.53) were much lower than nutrients (R2 0.62-0.64 and NS=0.90-0.93). Hence, the results presented in this manuscript is relevant and within the preview of the SWAT research community.

This is the first study of an area limited to the region having a large amount of strip-mined land. We believe this study will motivate researchers to collect more data and develop a more robust and better SWAT model in the future. We will be continuing this work and try to bring local partners to collaborate in the future works and disseminate the results that benefit all section of society and stakeholders in the Appalachian region or regions with dominant strip-mined lands.

We request the reviewer to consider these situations and let us know his views.

References:

Moriasi, D.N.; Gitau, M.W.; Pai, N.; Daggupati, P. Hydrologic And Water Quality Models: Performance Measures And Evaluation Criteria. Transactions of the ASABE 2015, 58, 1763-1785.

Feng, Q.; Chaubey, I.; Her, Y.G.; Cibin, R.; Engel, B.; Volenec, J.; Wang, X. Hydrologic and water quality impacts and biomass production potential on marginal land. Environmental Modelling & Software 2015, 72, 230-238

Love, B.J.; Nejadhashemi, A.P. Water quality impact assessment of large-scale biofuel crops expansion in agricultural regions of Michigan. Biomass & Bioenergy 2011, 35, 2200-2216
